# The adaptive value of density-dependent habitat specialization and social network centrality

Quinn M. R. Webber [1,3] ✉, Michel P. Laforge [2,4], Maegwin Bonar[2,5] & Eric Vander Wal[1,2]

Density dependence is a fundamental ecological process. In particular, animal habitat selection and social behavior often affect fitness in a density-dependent manner. The Ideal Free Distribution (IFD) and niche variation hypothesis (NVH) present distinct predictions associated with Optimal Foraging Theory about how the effect of habitat selection on fitness varies with population density. Using caribou (*Rangifer tarandus*) in Canada as a model system, we test competing hypotheses about how habitat specialization, social behavior, and annual reproductive success (co)vary across a population density gradient. Within a behavioral reaction norm framework, we estimate repeatability, behavioral plasticity, and covariance among social behavior and habitat selection to investigate the adaptive value of sociality and habitat selection. In support of NVH, but not the IFD, we find that at high density habitat specialists had higher annual reproductive success than generalists, but were also less social than generalists, suggesting the possibility that specialists were less social to avoid competition. Our study supports niche variation as a mechanism for density-dependent habitat specialization.

Our understanding of animal ecology can be incorporated into five fundamental principles: organisms consume resources, require space to live, interact with members of the same and other species, live in dynamic environments (including resources and other species), and copy their genes[1]. Each of these principles relates to population density. Animals consume resources as they are available, but as population density increases, resources become increasingly limited, and competition among conspecifics influences how animals use space, interact with conspecifics, and copy their genes. A salient question in the integration of these fundamental principles lies in disentangling apparent social behavior from shared preferences for habitats or resources and to assess the relative impacts of social behavior and habitat selection on individual fitness parameters, i.e., survival and reproductive success[2]. Patterns of habitat selection, i.e., the non-random use of available habitats[3], can vary based on the social environment an animal experiences, for example, an individual's own social phenotype and spatiotemporal variation in population density[4]. Importantly, individual variation in social phenotypes also can be density-dependent[5]. Understanding of the adaptive value of density-dependent habitat selection and social phenotypes influences our ability to quantify individual-based traits and assess their influence on fitness components.

Density dependence of phenotypes influences population dynamics and demographic rates through feedbacks between evolutionary (e.g., mean trait changes due to fitness differences between phenotypes) and ecological (e.g., population growth) processes[6,7]. An example of how density dependence of phenotypes affects population dynamics is through effects of a phenotype on survival or reproductive

[1]Cognitive and Behavioural Ecology Interdisciplinary Program, Memorial University of Newfoundland, St. John's, NF, Canada. [2]Department of Biology, Memorial University of Newfoundland, St. John's, NF, Canada. [3]Present address: Department of Integrative Biology, University of Guelph, Guelph, ON, Canada. [4]Present address: Department of Zoology and Physiology, University of Wyoming, Laramie, WY, USA. [5]Present address: Department of Ecology & Evolutionary Biology, Yale University, New Haven, CT, USA. ✉e-mail: qwebber@uoguelph.ca

success[8]. Density fluctuates in natural populations, suggesting that individuals should display behavioral plasticity in response to fine-scale spatiotemporal changes in population density[9]. For gregarious species, social network centrality (i.e., the extent to which an individual is socially connected to others)[10,11] and interaction duration[12] are density-dependent, and the relationship between these traits and fitness is predicted to change as a function of population density[13,14]. For example, if increasing density results in higher individual sociality, we might predict the most social individuals will have higher fitness. However, if increasing density reduces individual sociality as animals space apart to reduce competition, we might predict the most social individuals have lower fitness due to competition[11]. Individuals in social groups should therefore exhibit adaptive social plasticity to cope with density-dependent increases in 'apparent competition' (hereafter, competition) among conspecifics[15]. The adaptive value of social behavior and the potential for social plasticity in the context of density dependence is not often considered in studies focused on the adaptive value of sociality. While studies highlighting the link between sociality and density have become increasingly common[11,15,16], few empirical studies explicitly quantify individual sociality and fitness and assess how this relationship varies across a gradient of population density (but see ref. 17). As a result, there are few empirical examples that demonstrate the effect of population density on the sociality-fitness relationship. The relationship between social behavior and fitness has potential to influence, and be influenced by, population-level density dependence.

Animals typically cannot be social without sharing geographic and environment space and the habitats within these spaces. Habitat selection is also density-dependent and affects fitness[18-20]. Density-dependent habitat selection occurs when individuals select habitat based on habitat quality, modified by the density of individuals present[18]. Habitat selection analyses are used to predict how populations, or individuals, select certain habitats compared to their availability[21]. Habitat selection phenotypes vary among individuals[22], across densities[23], and infer foraging behavior[19]. Optimal Foraging Theory presents two distinct subsets of literature with associated predictions about how habitat, resource, and dietary selection and specialization have evolved as a function of variation in population density: Ideal Free Distribution and its counterpart the Ideal Despotic Distribution and the Niche Variation Hypothesis.

The Ideal Free Distribution (IFD) suggests the available resources in a habitat patch sustain a specific number of individuals, or carrying capacity. Animals distribute themselves among two or more habitat patches such that mean fitness in each habitat is equal, wherein habitats vary in their resources and the density of animal within the habitat[18,24]. Meanwhile, the Ideal Despotic Distribution (IDD) suggests that habitat selection by subordinate individuals is constrained by territoriality of dominant individuals[25]. Density-dependent habitat selection is an extension of IFD and IDD theory. IFD theory invokes an assumption that populations alter their habitat selection based on the relative profitability of two or more habitats with the assumption that habitat quality should decline with increasing population density[1,18]. Moreover, the mechanism driving density-dependent habitat selection assumes population density is a proxy for competition[1,19]. The role of competition in habitat selection is a critical aspect of density-dependent habitat selection, where increased density assumes increased competition and competition occurs in the form of fine-scale social interactions[21]. Social biologists have long cited competition for resources (e.g., mates or food) as one of the major costs of group living[26,27]. Behavioral ecologists assume competition is an inherently social process that occurs through auditory, olfactory, or visual intimidation or through physical aggression[28,29]. Given that density-dependent competition likely influences social behavior, we endeavor to disentangle the relative effects of social behavior and habitat selection on fitness[2].

Feeding competition determines spatial position of individual animals in foraging groups. For example, when a social group spreads out in space, individuals may not be feeding at the same fine-scale patch, thus increasing the per capita resource available to each individual[28]. Spacing out may also facilitate access to multiple patch types or food items. Within-group (or population) competition therefore has potential to shape individual foraging decisions and affect the degree to which individual animals generalize or specialize on resources or habitats[30]. In an ideal free scenario, animals forage in different habitats and deplete food resources proportional to population density in each habitat. In theory, assortment of animals into discrete habitat classes should minimize, or at least stabilize, feeding competition[18]. Feeding competition is implicit within the IFD as a likely mechanism that drives variation in the relationship between population density, habitat selection, sociality, and foraging specialization.

The IFD predicts that individuals at high population density will be generalist consumers because competition for high quality resources is high, while at low population density individuals will be specialist consumers[31]. For example, grasslands are considered high quality habitat for red deer (*Cervus elaphus*). Deer were grassland specialists at low density but habitat and diet generalists, at high density[32]. The Niche Variation Hypothesis (NVH)[33] posits an increase in between-individual variation in resource use. The degree to which individuals vary in their resource use falls along a generalist–specialist continuum, where the proportion of an individual's diet relative to the population's overall resource base defines the extent to which an individual specializes on available resources[34]. Individuals vary in their resource use to reduce intraspecific competition through specializing on a subset of the resource available to the population[34-36]. The NVH predicts that individuals should become resource specialists when population density is high[34]. For example, individual banded mongoose (*Mungos mungo*) increased their foraging specialization as group size and competition increased[30]. Notably, both the IFD and NVH assume optimality, but how optimality partitions among individuals is a function of changing population density. Given these diverging predictions about habitat specialization it is also possible that individuals may display plasticity in their ability to specialize within their lifetime[34].

Plasticity is the variation in a given trait, including social and behavioral traits, as a function of variation in internal or external stimuli[37]. Within-individual behavioral plasticity, or flexibility, refers to the extent to which an individual's behavior changes in different situations or in response to a given stimulus, and this type of behavioral plasticity has been widely applied to the field of animal personality[37]. Animal personality traits, defined as consistent individual differences in behavior, are expected to persist through space and time and this variation may be adaptive[38]. The concept of individual differences in behavior can be interpreted and quantified as three components. (1) Behavioral plasticity: the ability of individuals to alter phenotypes as a function of the environment[37]. (2) Behavioral syndromes: correlated suites of behaviors across time or space[39]. (3) Behavioral repeatability: the proportion of phenotypic variance attributable to among-individual differences[40]. The integration of individual differences in social behavior and habitat specialization (and the associated components: plasticity, behavioral syndromes, and repeatability) remains a challenge given the difficulty in simultaneously measuring social and spatial behaviors[4] and here we address the challenge by using GPS relocations to estimate these behaviors for individual animals.

We empirically quantified social associations, habitat specialization, and fitness in six herds of a caribou (*Rangifer tarandus*) living across a population density gradient through space and time. Like most ungulates, caribou also have low variance in their adult survival (upwards of 90% of adults survive a given year in our system[41]), but high variance in offspring survival suggesting that fitness effects would

more likely be detected in annual reproductive success[42–44]. Caribou are social ungulates that live in fission-fusion societies[45] and at broader scales conform to the ideal free distribution (e.g., during calving[46]). We view the IFD as a null model when considering animal space use. Assumptions of the IFD include cost free movement between patches and perfect knowledge of habitat[18]. Notably, potential violations mark important contributions in that the violation of an assumption may represent an explanation for outcomes that deviate from the null IFD model. For example, among relatives there is no free choice, suggesting that individuals maximizing inclusive fitness may overexploit habitats at a given density, even though classic IFD theory suggests they should re-assort to maximize fitness[47]. Moreover, social animals are not necessarily free to move among habitats in an ideal way as the benefits of social foraging are predicted to outweigh the costs[48]. These assumptions of the IFD have historically yielded novel insight into how space use of grouping animals reinforces social behaviors.

To test the IFD and NVH, we first used proximity-based social network analysis to estimate social graph strength for individual caribou, which is the sum of weighted associations in a social network. Second, we estimated individual habitat specialization, measured as the proportional similarity in resource use between individuals and the population. Third, we estimated fitness based on annual reproductive success, an important fitness proxy in ungulates[43]. We then used multivariate behavioral reaction norms (BRNs) to estimate plasticity of social strength and habitat specialization across a population density gradient, covariance between social strength, habitat specialization, and annual reproductive success, and repeatability of all traits. We first tested predictions associated with socioecological theory, the IFD, and the NVH (for details on each prediction see Table 1). Note, we do not include predictions associated with the IDD because caribou are non-territorial and typically do not defend resources. We predicted that individual values of social strength should increase with population density based on the expectation that higher density increases the probability of interaction[49] (P1). According to the IFD and NVH, the relationship between habitat specialization and population density should differ, such that the IFD predicts individuals should generalize at high density (P2a), while the NVH predicts individuals should specialize at high density (P2b). We did not expect the relationship between social strength and habitat specialization to vary between the IFD and NVH. Under both scenarios, we predicted a positive relationship, such that more socially connected individuals are habitat generalists because competition among social groups should be lower if

individuals have generalist strategies (P3). Finally, we predicted that at lower density, annual reproductive success would be highest for individuals with a high degree of habitat specialization, while at higher density, annual reproductive success would be highest for individuals with a high degree of habitat generalization (P4a). By contrast, based on the NVH, we predicted that at lower density, annual reproductive success would be highest for individuals with a high degree of habitat generalization, while at higher density, annual reproductive success would be highest for individuals with a high degree of habitat specialization (P4b).

In addition to the predictions associated with IFD and NVH (see above), we also sought to quantify repeatability (r) of social strength and habitat specialization. Behavioral traits are typically considered highly repeatable if $r > 0.40$, moderately repeatable if $0.20 > r < 0.40$, and low or negligible repeatability if $r < 0.20$[40]. Notably, our objective was to estimate repeatability to place social strength and habitat specialization within a broader behavioral and evolutionary ecology context (objective 1 listed in Table 1)[50,51]. Finally, the IFD does not explicitly incorporate inter-individual behavioral variation. Our objective was to explore the relationship between annual reproductive success and sociality across density through the lens of the IFD. We might expect that at lower density, annual reproductive success would be highest for more socially connected individuals due to reduced competition at low density, while at higher density (and therefore higher competition) annual reproductive success would be highest for less social individuals that seek to avoid competition (Objective 2). For details on all predictions and objectives see Table 1.

## Results
### Summary
We monitored behavior and population dynamics for 127 individual adult female caribou in six herds over seven years (Fig. S1). In total, we calculated an average of $6.0 \pm 3.5$ (range: 1–14) measures of social strength, habitat specialization, and reproductive success per individual, for a total of 752 measures of these variables across all years, seasons, and herds. Due to variation in length of time that collars were deployed on individuals, seasonal networks were larger in winter (average: $66 \pm 21$ individuals, range = 35–90) than during calving (average: $53 \pm 26$ individuals, range = 15–81). On average, social strength was higher in winter (mean = $0.012 \pm 0.001$) than calving (average: $0.005 \pm 0.006$). Average habitat specialization indices were the same in winter (average: $0.72 \pm 0.08$) and calving (average:

**Table 1 | Summary of predictions**

| General prediction | Prediction associated with Ideal Free Distribution | Prediction associated with Niche Variation Hypothesis | Reference figure or table |
|---|---|---|---|
| P1: Density-dependent social strength. As density increases, individuals are expected to increase their social network strength. | P1a: No directional prediction. | P1b: No directional prediction. | Fig. 1a |
| P2: Density-dependent habitat generalization at high density (IFD) or habitat specialization at high density (Niche Variation Hypothesis). | P2a: As density increases, individuals are expected to become habitat generalists[31]. | P2b: As density increases, individuals are expected to become habitat specialists. | Fig. 1b |
| P3: Phenotypic covariance between social strength and habitat specialization[2]. | P3: More social individuals are expected to be habitat generalists. | Fig. 2 | |
| P4: Adaptive value of density-dependent habitat specialization. | P4a: Low density: higher fitness for habitat specialists. High density: higher fitness for habitat generalists[31,32]. | P4b: Low density: higher fitness for habitat generalists. High density: higher fitness for habitat specialists[36]. | Fig. 3 |
| Objective 1: Repeatability of social strength and habitat specialization, such that behavioral traits are expected to be consistent through space and time[40]. | No expectation. Traits are typically considered highly repeatable if $r > 0.40$, moderately repeatable if $0.20 > r < 0.40$, and low or negligible repeatability of $r < 0.20$. | | Table 3 |
| Objective 2: Adaptive value of density-dependent social strength[2]. | Low density: higher fitness for more social individuals High density: higher fitness for less social individuals. | - | Table 3 |

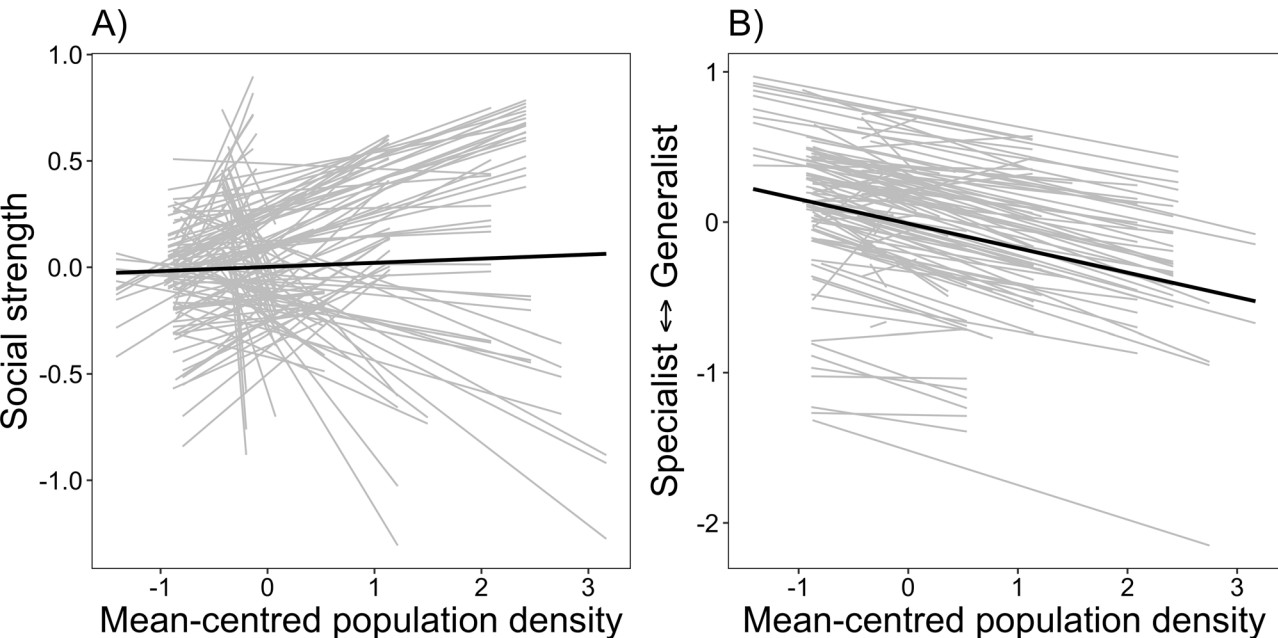

**Fig. 1 | Social strength and habitat specialization behavioural reaction norms.** Behavioral reaction norms testing the relationship between mean-centered population density and (**A**) social network strength and (**B**) habitat specialization for caribou (*Rangifer tarandus*; *n* = 127) in Newfoundland. Note, both social network strength and habitat specialization are presented as best linear unbiased predictors (BLUPs) extracted from Bayesian mixed models. Each line represents an individual behavioral response to changes in population density and crossing of lines represents individual differences in plasticity (i.e., an individual-environment interaction). Source data are provided as a Source Data file.

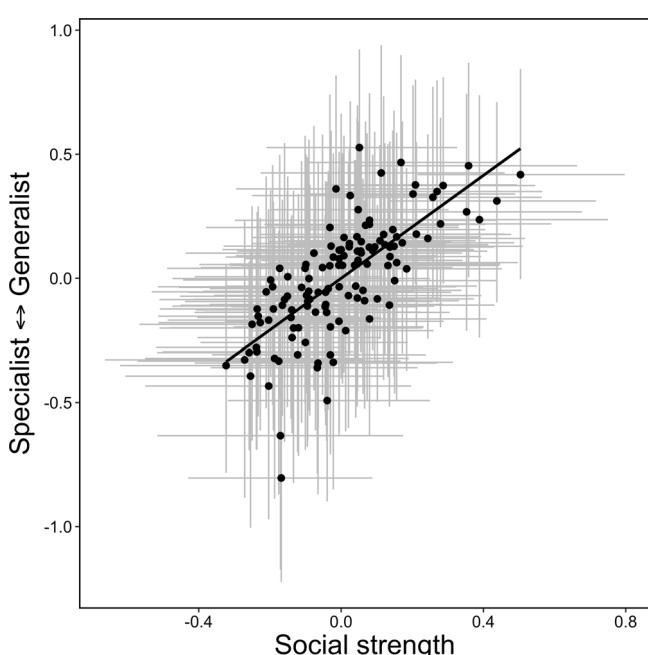

**Fig. 2 | phenotypic covariance between social strength and habitat specialization in caribou (*Rangifer tarandus*, *n* = 127) in Newfoundland.** Note, both variables are extracted from best linear unbiased predictors (BLUPs) extracted from Bayesian mixed models for visualization and gray lines represent 95% credible intervals around BLUPs for social strength (horizontal lines) and habitat specialization (vertical lines). Source data are provided as a Source Data file.

0.72 ± 0.13). Habitat specialization was positively correlated with habitat selection coefficients generated from resource selection functions for the four most common habitat types. Given that the $PS_i$ measures specialization of a given resource relative to the population, a positive relationship between selection and specialization suggests that specialists tend to select for a single habitat type and neither select nor avoid other available habitat types., while generalists neither selected nor avoided all habitat types (Tables S2; S3; Fig. S6). Because most caribou have strong selection for lichen, there were few, if any, caribou that specialized on lichen (Fig. S6), whereas some individuals specialized on, and had strong selection for, other habitat types. With regards to fitness, calf survival was 61% (241/393 annual reproductive events) over the course of our study.

### Density-dependent phenotypes

We found support for our first hypothesis that social strength and habitat specialization would increase as a function of population density gradient (Predictions 1 and 2). Individuals varied their behavioral response to changes in population density, such that some individuals became less social as population density increased, but most individuals were more socially connected as density increased (P1, Fig. 1a, Fig. S9). In addition, individuals also varied their habitat selection patterns as population density changed, where most individuals tended to become habitat specialists as density increased (P2a, Fig. 1b, Fig. S9). Although the direction of behavioral change in habitat specialization was similar for most individuals, we observed variation in the magnitude of change, suggesting an individual by environmental interaction.

### Phenotypic covariance

We found mixed support for predictions on phenotypic covariance (P3) and repeatability (P4). In our global model, we found strong phenotypic covariance between social strength and habitat specialization (0.52, 95% Credible Interval: 0.21, 0.79), suggesting that habitat generalists were more socially connected and habitat specialists were less social (Fig. 2). After taking herd, season, and year into account as fixed effects, we found that social strength was moderately repeatable during calving ($r$ = 0.25, 95% CI: 0.15, 0.37), but not winter ($r$ = 0.03, 95% CI: 0.015, 0.05). By contrast, habitat specialization was moderately repeatable in winter ($r$ = 0.20, 95% CI: 0.11, 0.29), but not during calving ($r$ = 0.09, 95% CI: 0.05, 0.14, Table 2).

When testing the relationship among social strength, habitat specialization, and fitness, we found support for the NVH. In our global model, there was a positive relationship between habitat specialization and social strength, where more socially connected individuals were habitat generalists (P3, 0.50, 95% CI: 0.17, 0.71, Table 3). In our global model, there was a weak negative relationship between habitat specialization and fitness (−0.29, 95% CI: −0.59, 0.03, Fig. 3), but no relationship between social strength and fitness (−0.03, 95% CI: −0.36, 0.29, Table 3, Fig. S7). When we modeled high and low density separately, there was no effect of social strength on fitness at either low or high density (P5a and P5b, Table 3). In partial support of the NVH (P6b), and in contrast to the IFD (P6a), we found negative covariance between habitat specialization and fitness at high density (−0.62, 95% CI: −0.99, −0.01, Table 3), such that habitat specialists had higher fitness at high density and habitat generalists and low fitness at high density. By contrast, the habitat specialization-generalization continuum had no effect on fitness at low density (0.02, 95% CI: −0.81, 0.94, Table 3).

## Discussion

Animals live by five fundamental principles that are distilled into resources, space use, competition, environmental variation, and reproduction[1]. We examined these principles by testing competing hypotheses about the relationships among habitat specialization, sociality, population density, and fitness. According to the IFD, resource specialists maximize fitness at low population density and generalists at high density[31], while niche variation posits that resource specialists maximize fitness at high population density[36]. The apparent tension between these two hypotheses may be mediated by considering the social environment experienced by individuals[30]. An increase in social connections across a population density gradient could influence individuals' propensity to successfully generalize or specialize. At high density, when individuals tend to be more socially connected and compete more for limited resources, individuals may benefit more from specializing on different available resources to reduce competition[52]. Here, we highlight that individual habitat specialization is density-dependent following predictions associated with the NVH, and the relationship between habitat specialization and fitness is moderated by individual social phenotypes.

Fretwell & Lucas[18] proposed the IFD as a null model and our findings reinforce previously published studies on red deer[32], roe deer (*Capreolus capreolus*)[53], and birds[54,55]. Overall, we found support for our predictions associated with the NVH, where individuals tended to specialize on one habitat at high population density (P4b). In banded mongooses, sea otters (*Enhydra lutris*), and stickleback (*Gasterosteus aculeatus*), individuals and populations tended to specialize at high population densities[30,36,56]. In addition to these empirical studies, our results support theory suggesting that population density is a mechanism driving variation in individual habitat specialization[34]. The relationship between habitat specialization and fitness according to the NVH assumes that individuals specialize on profitable resources and that this profitability results in increased fitness. Indeed, we found that higher fitness was achieved for habitat specialists at high density. Given that individuals consistently adjusted their habitat specialization behavior as density changed, and that at high densities specialists had higher fitness, fluctuating selection should favor individual variation in habitat specialization. A potential mechanism explaining among-individual variation in habitat specialization is a mutual interest in avoiding competition in heterogeneous or patchy environments[57]. Given the adaptive value of habitat specialization, plasticity in habitat specialization from low to high density could be maintained as individuals alter their behavior to adjust to environmental conditions.

In support of our prediction, we found positive phenotypic covariance between social strength and habitat specialization, such that more socially connected individuals were habitat generalists (P3, Table 3). Social behavior and habitat selection occur in the same geographical space, i.e., animals must share space to interact. Because the process of habitat specialization and the drivers of social behavior are intertwined, animals might gain fitness benefits from their shared social connections, while they may also gain fitness benefits from habitat specialization[53]. Individual resource specialization is driven by competition[34]. For example, density-mediated competition for preferred prey is the likely driver of dietary variation in sea otters[58]. In a more competitive social environment, IFD theory predicts that individuals should generalize on resources or habitats to reduce competition[31]. Social individuals may be constrained from specializing due to social connections and the competition associated with group living at high density. Moreover, theory of density dependence predicts that at high population density, reproductive success will be relatively low[59], and only a small proportion of individuals will successfully rear calves. Habitat generalists tend to be more socially connected – a tactic that does not immediately affect fitness. More social habitat generalists presumably obtain other benefits of group-living, such as increased vigilance or access to information about foraging resources. Although we were unable to test for life-history trade-offs, it is possible more socially connected adults have a higher probability of adult survival and therefore face a trade-off between survival and reproductive success that could have implications for population dynamics. Moreover, we were unable to infer male fitness. Given observed plasticity in social behavior and habitat specialization, these contrasting strategies present an apparent tension for individuals to simultaneously be habitat specialists *and* be highly connected

**Table 2 | Summary of repeatability (r) estimates for caribou social strength and habitat specialization**

| Trait | Season | Median (±SD) | Repeatability | V_res |
|---|---|---|---|---|
| Social strength | Calving | 0.005 ± 0.006 | 0.25 (0.15, 0.37) | 1.54 |
| | Winter | 0.012 ± 0.015 | 0.028 (0.015, 0.05) | 0.15 |
| Habitat specialization | Calving | 0.72 ± 0.13 | 0.09 (0.04, 0.14) | 1.07 |
| | Winter | 0.72 ± 0.08 | 0.20 (0.11, 0.29) | 0.44 |

Repeatability measures are a ratio between the proportion between-individual variance attributable to the residual variance (Vres) and therefore does not go below zero. High repeatability values are typically values are >0.4, moderate values of repeatability are between 0.2 and 0.4, and low values of repeatability are <0.20. Values in brackets represent 95% credible intervals extracted from MCMC models.

**Table 3 | Phenotypic covariance among behavioral reaction norm intercepts for social strength, habitat specialization, and fitness in models with all data and separated into separate datasets where only data in the lowest 25% quantile, and highest 75% quantile, of population density were included**

| Trait combination | All data | Low density (25% quantile) | High density (75% quantile) |
|---|---|---|---|
| Social strength, habitat specialization | **0.50 (0.17, 0.78)** | – | – |
| Social strength, fitness | −0.03 (−0.36, 0.29) | −0.34 (−0.99, 0.86) | 0.40 (−0.84, 0.99) |
| Habitat specialization, fitness | −0.29 (−0.59, 0.03) | 0.02 (−0.81, 0.94) | **−0.62 (−0.99, −0.01)** |

Numbers in brackets are 95% credible intervals and phenotypic covariance is considered significant if credible intervals do not overlap zero.
Bold values represent covariance estimates where 95% credible intervals do not overlap zero.

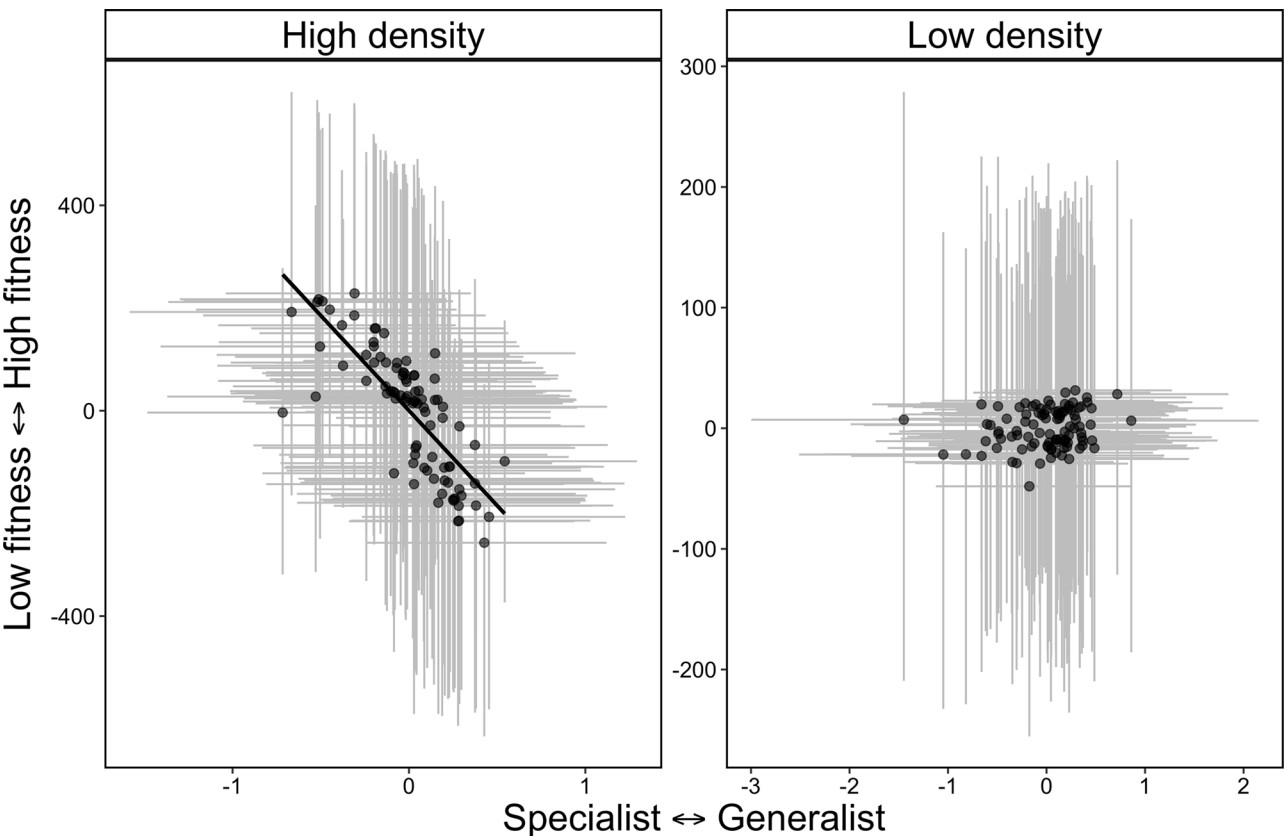

**Fig. 3 | phenotypic covariance between reproductive success and habitat specialization at relatively high (left panel) and relatively low (right panel) population density for caribou (*Rangifer tarandus*, *n* = 127) in Newfoundland.** At high density, more specialized individuals also tended to have an overall higher fitness value, whereas there was no effect of habitat specialization on fitness at low density. Note, both variables are extracted from best linear unbiased predictors (BLUPs) extracted from Bayesian mixed models for visualization and gray lines represent 95% credible intervals around BLUPs for habitat specialization (horizontal lines) and fitness (vertical lines). Source data are provided as a Source Data file.

in the social network; an outcome which might be the driver of plasticity.

Our integration of individual habitat specialization within a behavioral reaction norm framework highlights the ability for individuals to adjust their specialization phenotypes across a population density gradient. While plasticity in morphological traits is known to influence dietary specialization for Eurasian perch (*Perca fluviatilis*)[60], plasticity of habitat specialization is less well understood. Despite relatively few empirical studies, plasticity in individual specialization reflects an extension from the expectations of the NVH, which posits contrary predictions to the IFD. Individuals that experience a range of population densities within their lifetime should vary in their habitat specialization-generalization phenotype across densities[9]. We found that individual caribou generally became more specialized as population density increased, suggesting within-individual plasticity – a strategy that represents an individual's ability to acclimate to changing environmental conditions. Since reproductive success is frequently depressed at high density[61,62], our results suggest that the most specialized individuals have highest reproductive success, although it is possible that other ecological or behavioral factors could influence reproductive success. The ability for individuals to modulate their specialization behavior across population densities therefore likely has adaptive consequences[63].

Consistent with results from a recent meta-analysis of spatial phenotypes[64], we found that habitat specialization was moderately repeatable, suggesting that the most specialized individuals at low population densities remain the most specialized at a higher density. Similarly, in bottlenose dolphins (*Tursiops aduncus*), the same measure of habitat specialization (the proportional similarity index) was repeatable through time[65]. Behavioral repeatability is important in an evolutionary context because repeatability represents the upper limit of heritability[66], and ultimately, the adaptive value of habitat specialization suggests the potential for this trait to undergo natural selection[64].

Our study focused on adult female caribou. It is possible that additional data from males may alter our findings. However, in spring and summer, females tend to form small nursery groups comprised exclusively of females and newborn calves, while adult males tend to be solitary in summer[67]. Meanwhile, during winter, caribou tend to form larger mixed-sex groups[67]. We acknowledge the limitation of only including adult females, however, it is worth noting that the drivers of fitness for male caribou likely vary given there is no parental care and males tend to be less social than females[67]. Moreover, our proxy for fitness was calf survival, as opposed to adult survival. In ungulates, once animals are recruited into the population, survival is very high (i.e., typically >90% per year), while offspring survival prior to recruitment is highly variable[42,44]. Variation in offspring survival has various causes, including climate induced changes in vegetation availability and predation[68] as well as foraging related reductions in nutrition[69]. In our study, cause of mortality for individual calves was unknown, but the link between nutrition, foraging specialization, and calf survival exists[69] and we propose future work address these relationships to further elucidate the effect of maternal nutrition on calf survival in caribou and other ungulates.

Animals use space, select habitat, and occupy social positions that are intended to maximize their fitness. By integrating distinct

components of density-dependent Optimal Foraging Theory with competing hypotheses derived from the Ideal Free Distribution and the Niche Variation Hypothesis, we test the effects of social and spatial phenotypes as drivers of fitness. We present evidence supporting predictions of the NVH that highlight the adaptive value of individual habitat specialization was high at high population density. The adaptive value of habitat specialization across a population density gradient has implicit implications for our understanding of behavioral eco-evolutionary dynamics[13,14]. While we do not explicitly test for eco-evolutionary dynamics, our study addresses two of the criteria outlined as prerequisites for eco-evolutionary dynamics[6]. First, previous work in this system has identified fluctuations in population density through time[70] and although we only included data from seven years, we observed differences in the distribution of habitat specialization as a function of population density. Second, we identified an effect of habitat specialization on fitness at high, but not low, density. Furthermore, if the adaptive value of a trait varies with density and population growth, this provides some evidence for eco-evolutionary dynamics[71]. Although estimating eco-evolutionary dynamics for behavior remains elusive, we satisfy some of the baseline expectations of an eco-evolutionary correlation. Next steps include identifying a plausible mechanistic link between an evolutionary (e.g., mean trait changes due to fitness differences between phenotypes) and ecological (e.g., population growth) process[6]. Density dependence is a fundamental ecological process, and we highlight the effects of population density on the relationship between spatial and social behavioral phenotypes and fitness.

## Methods

### Study area and species
All animal capture and handling procedures were consistent with the American Society of Mammalogists guidelines[72] and permits were not required for the following work as all data collection was conducted by government agencies responsible for permitting. We used global positioning system (GPS) location data collected from six caribou herds in Newfoundland, Canada (Fig. S1; Supplementary Note 1). Caribou population density in Newfoundland has fluctuated through time (Fig. S2). Herds peaked in size in the 1990s and declined in the 2000s[70]. Adult female caribou from all herds were immobilized and fitted with GPS collars (Lotek Wireless Inc., Newmarket, ON, Canada, GPS4400M collars, 1250 g, see Supplementary Note 1 for details). Collars were deployed on 127 adult female caribou for one to three years, and collars were often re-deployed on the same individuals for up to seven years (mean ± SD = 3.2 ± 1.7) between 2007 and 2013. The number of collared individuals varied between herds, but the proportion of collared individuals in each herd was similar (Fig. S4). Collars were programmed to collect locations every two hours. Prior to analyses, we removed all erroneous and outlier GPS fixes following[73]. We assigned each relocation a habitat classification extracted from Landsat images with 30 × 30 m pixels. To assess potential for seasonal differences in social behavior and habitat selection, we delineated GPS fixes into discrete 70-day periods to reflect winter (1 December–10 February) and calving (21 May–31 July). Seasons correspond with previously identified seasonal periods that were identified based on caribou movement and life-history[74]. Caribou are pregnant through winter when food availability is low. Adult female caribou form groups to optimize access to foraging resources[67], while females employ a capital breeding strategy that relies on maternal stores for fetal growth[75]. Nutritional demands of reproduction peak during lactation and caribou finance these costs through dietary income, as opposed to stored fat (i.e., capital income:[76]).

### Population density estimates
Population size was estimated based on aerial surveys for each herd (Fig. S2)[77]. The majority of herds in Newfoundland have been monitored and surveyed intermittently and opportunistically between 1979 and 2014[78]. Between 2009 and 2013, herds were surveyed in spring or winter by an observation crew that typically consisted of at least two observers, but pilots also occasionally acted as secondary observers. Aerial surveys were primarily flown in rotary-wing aircrafts and were conducted either opportunistically by locating caribou and counting groups or by using traditional aerial surveys in a systematic strip, random block, stratified-random block or mark-resight design[78]. We estimated the area occupied by each herd in each season and year by pooling GPS relocation data for all individuals and subsequently calculating the area of the 100% minimum convex polygon in the *adehabitatHR* (version 0.4.21) package in R[79]. We then estimated population density for each herd in each year and season by dividing the total number of animals estimated by the area occupied by the herd. To ensure convergence of subsequent models, population density was scaled and mean centered by herd to preserve variation in density among herds.

### Social network analysis
We generated proximity-based social networks from GPS telemetry data. Traditional designation of caribou herds in Newfoundland assigns animals to specific herds, however, because of winter spatial overlap for some herds[80], we constructed a single network for all collared animals in each year-by-season combination. We generated social networks based on proximity of GPS fixes for individual caribou. We assumed association between two individuals if simultaneous GPS fixes, i.e., recorded within 5 min of each other, were within 50 m of one another[45,81]. We applied the 'chain rule', where each discrete spatio-temporal GPS fix was buffered by 50 m and we considered individuals in the same group if 50 m buffers for two or more individuals were contiguous[82]. We weighted edges of social networks by the strength of association between dyads using the simple ratio index (SRI, see Supplementary Note 2). The SRI is a shared dyadic value that measures the number of times the dyad were observed together, while accounting for the amount of data for each individual[83]. All social networks were generated using the *spatsoc* (version 0.2.2) package[84] in R.

Given recent discussion regarding the use of effect sizes and Bayesian inference to model social networks[85], we did not generate null models and estimate effects of covariates on social network strength in a multi-variate regression framework. The social network null modeling framework proposes data-stream permutations should be conducted to ensure observed measures of sociality are non-random. In many cases, common data stream permutations are not appropriate for hypothesis testing when using regression models[86]. Specifically, data stream permutations can result in extremely high type I error rates and the null expectation of a random network is not appropriate in most systems[86,87]. Moreover, when using Bayesian regression models, the traditional data stream to model coefficient pipeline[86,88] is not appropriate due to the iterative nature of Bayesian statistics[87,89]. However, despite concerns with social network permutation techniques, there remains an expectation to ensure observed relationships are indeed non-random. We therefore followed past work in our system[81,90] and developed a parallel set of univariate frequentist models and developed data-stream permutations to assess whether the relationships between social graph strength and covariates were non-random (Supplementary Note 2; Figs. S4 and S5). We generated null models based on GPS fixes (i.e., data stream permutations) to reduce potential for type II error typically associated with node-based permutations (Supplementary Note 2)[91].

### Estimating habitat specialization
Our study area was separated into eight habitat types based on landcover classification: conifer forest, conifer scrub, mixed-wood forest, deciduous forest, wetland, lichen barrens, rocky

barrens, and water/ice[92]. The proportional similarity ($PS_i$) is a measure of interspecific dietary overlap and evenness which accounts for the amount of a resource an individual consumes relative to the population[93]. Using the number of spatial relocations for each individual in each habitat type, we estimated the $PS_i$:

$$PS_i = 1 - 0.5 \sum_j |p_{ij} - q_j| \qquad (1)$$

where $p_{ij}$ describes the proportion of the $j$th habitat type for individual $i$, and $q_j$ describes the proportion of the $j$th habitat type at the population level. Values of $PS_i$ closer to one reflect individuals that select habitats in direct proportion to the population, i.e., habitat generalists, whereas values of $PS_i$ closer to zero reflect individuals that are habitat specialists. We calculated the $PS_i$ using the *RInSp* (version 1.2.4) package in R[94]. A value of $PS_i$ was calculated for each individual in each year-by-season combination and represented the degree to which that individual specialized on any given habitat type. To confirm habitat specialization was related to habitat selection, we generated resource selection functions[95] and compared the $PS_i$ to habitat selection coefficients for the dominant habitat types (see Supplementary Note 3, Fig. S6).

## Fitness estimates

We used annual reproductive success as a proxy for fitness for adult female caribou. Caribou only have a single calf per year. Parturition is associated with reduced movement rate in caribou, and we used inter-fix step length from GPS collared caribou to infer parturition and calf mortality[46,96]. To measure calf survival, we applied a population-based method using a moving window approach to evaluate three-day average movement rates of adult females to estimate parturition status[97], and an individual-based method that used maximum likelihood estimation and GPS inter-fix step length of adult females to estimate calf mortality up to four weeks in age. Mothers that do not give birth have a consistent daily average movement through time, while mothers that give birth decrease step length immediately after birth and slowly return to daily average movement rates[96]. In cases where calf mortality occurs, the mother will return to daily average movement rate almost immediately after calf mortality[96]. The majority of calf mortality in our study was due to predation from coyotes (*Canis latrans*) and black bears (*Ursus americanus*)[98,99]. Based on results from these models, we estimated annual reproductive success for each individual caribou in each year as a proxy for fitness.

## Statistical analysis: behavioral reaction norms

Behavioral reaction norms (BRNs) estimate behavioral repeatability and plasticity. BRNs generate three key parameters: (1) the reaction norm slope, which corresponds to phenotypic plasticity; (2) phenotypic covariance, which corresponds to behavioral syndromes; and (3) the reaction norm intercept, which corresponds to consistent individual differences in behavior, which are used to estimate repeatability. We employed multivariable mixed model to quantify BRN components, i.e., repeatability and plasticity, for resource specialization, social strength, and fitness as a function of population density. Despite criticisms of Bayesian models[100], we used multi-variable Bayesian models to avoid the common problem of 'stats-on-stats', where best linear unbiased predictors (BLUPs) are extracted from one or more mixed models and used to represent an individual's phenotype in subsequent statistical models[101,102]. BLUPs can be problematic if used in the context of 'stats-on-stats', i.e., the integration of an output from one model into another model, because each individual BLUP contains a measure of error, which increases Type I error (false positive) in the new model and acts as a confounding variable in the new model[102]. To facilitate model convergence, we scaled and zero-centered social

strength and habitat specialization to a mean of zero (see below for further details on convergence).

We developed five multi-variable models using the MCMCglmm (version 2.32) in R[103]. First, we parameterized a tri-variate global model that included calf survival ($f$), social strength ($s$), and habitat specialization ($h$) as co-response variables. We can visualize a simplification of our model structure as:

$$(f, s, h) \sim \beta_0 + \beta_1 f + \beta_1 s + \beta_1 h + \ldots + \varepsilon \qquad (2)$$

where the fixed effects (β) are estimated for each co-response variable (*f, s, h*). Meanwhile, modeling multiple response variables estimates covariance between random intercepts and slopes[104,105]. In our tri-variate model, we included year, season, scaled population density, and herd as fixed effects. Individual identity and mean and center-scaled population density were included as random effects, where individual values of social strength and habitat specialization varied as a function of population density. Next, we parameterized four bi-variate models with calf survival and either social strength or habitat specialization as co-response variables for subsets of the data delineated based on either low- or high-density herds (see Supplementary Note 4). Specifically, based on the distribution of scaled population density, we delineated the lowest quartile (lowest 25% of population density values) as low density data, and the highest quartile (the highest 75% of population density values) as high density data. We chose to separate data based on the lowest 25% and highest 75% values of population density to ensure there was no potential for error in assigning individuals to a density category or overlap of individuals in each herd.

Using results from the global model, we evaluated repeatability ($r$) of BRN intercepts for habitat specialization and social strength as the amount of between-individual variance ($V_{ind}$) attributable to the residual variance among groups ($V_{res}$) for each trait[106]:

$$r = \frac{V_{ind}}{(V_{ind} + V_{res})} \qquad (3)$$

Within the global model, repeatability was estimated for social strength and habitat specialization during winter and calving seasons. We also examined correlations between habitat specialization, social strength, and fitness. Among-individual variance in resource specialization and social strength may differ based on whether population density is low or high, relative to the overall average. We therefore varied residuals in the model by season because of differences in social tendencies and habitat selection for caribou across seasons[74,90]. Thus, we calculated $V_{res}$ and $r$ for habitat specialization and social strength, for each season separately. Finally, we used uninformative priors and coded variance ($s^2$) as $s^2/2$ and degree of belief as four for fixed and random effects. We fitted all models with Gaussian error structure for response variables. We ran all models for 420,000 iterations, a thinning length of 100, and a burn-in of 20,000 to form posterior distributions. The importance of fixed and random effects was judged by the distance of the mode of the posterior distribution from zero, and the spread of the 95% credible intervals. We evaluated model convergence by visually investigating chains, assessing the Heidelberger convergence diagnostic[107], and checking that auto-correlation between successive samples of the MCMC chain was below 0.1. Model convergence is the process of iteratively training a model until it can no longer improve its performance[108]. Finally, we performed three runs of our model to ensure different chains reached the same qualitative result. All analyses were conducted in R version 4.0.2[109].

## Reporting summary

Further information on research design is available in the Nature Portfolio Reporting Summary linked to this article.

## Data availability

The output data generated in this study have been deposited in Zenodo (https://zenodo.org/records/10903837). The raw GPS data are available under restricted access because of the sensitivity of caribou calving areas and access can be obtained by contacting the corresponding author. Source data are provided with this paper.

## Code availability

All code are available at: https://zenodo.org/records/10903837

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

## Acknowledgements

We respectfully acknowledge the territory in which data were collected and analyzed as the ancestral homelands of the Beothuk and the Island of Newfoundland as the ancestral homelands of the Mi'kmaq and Beothuk. We thank G. Albery and members of the Wildlife Evolutionary Ecology Lab for helpful comments on previous versions of this manuscript. We also thank members of the Newfoundland and Labrador Wildlife Division, including S. Moores, B. Adams, W. Barney, and J. Neville, for facilitating animal captures and for logistical support in the field. We thank T. Bergerud and S. Mahoney for their vision in initiating much of the work on caribou in Newfoundland. Funding for this study was provided by a Natural Sciences and Engineering Research Council (NSERC) Vanier Canada Graduate Scholarship to Q.M.R.W., NSERC Canada Graduate Scholarships to M.P.L. and M.B., and a NSERC Discovery Grant to E.V.W.

## Author contributions

Q.M.R.W. and E.V.W. conceived the idea and analytical framework, M.B. quantified reproductive success, M.P.L. provided guidance on spatial analyses, Q.M.R.W. conducted all analyses and wrote the manuscript, and all authors critically revised the manuscript.

## Competing interests

The authors declare no competing interests.
