## [Peer Review File · Nature Communications]

The adaptive value of density-dependent habitat specialization and social network centralityREVIEWER COMMENTS

Reviewer #1 (Remarks to the Author):

Please see my comments in the pdf attached.

Comments on manuscript NCOMMS-23-00651-T

The study presented in the manuscript tested the predictions generated based on the ideal free distribution (IFD) and the niche variation hypothesis (NVH) on how habitat specialization, social association strength and calf survival of 6 female caribou (*Rangifer tarandus*) herds covary under different spatiotemporal population densities. It has been shown that both habitat specialization and social association strength increase with population density, while habitat specialists are less social than generalists, thus supporting the NVH which posits that individuals should be more specialized as population density increases. Besides, it has been found that habitat specialization or generalization had no effect on fitness at low population density. The manuscript was well written and is novel to read — it tries to bring the IFD, the NVH, social behaviours, and their fitness consequences together to generate new insights. After reading, I'd like to feedback on a few things that might be worthwhile to think about.

The first concern I'd raise relates to the descriptions of the predictions by the hypotheses. In particular, the manuscript lacks mechanistic explanations on what led to the different predictions by the NVH and the IFD following the state-of-the-art discussions on them, and how the predictions specifically in this study were made based on a thorough review of these hypotheses. This also relates to the clarification on what the specific objectives the manuscript is aiming to achieve — what the study contributes to the literature in relation to the existing theory on IFD, NVH and social behaviour in particular. From my understanding, both the IFD and the NVH generate predictions on density and specializations, while the present study also incorporates the social aspects; however, the introduction did not clearly link these topics together with their own arguments, but keep generating predictions to test. I'd thus suggest more explanations on how these predictions were generated based on existing studies.

Unclear also remains in the method section. For example, it was unclear why only females were tracked and studied (and this left readers wondering how reliable the conclusions are when they are drawn from a completely female-biased

dataset), and how reliable those collared adult females as ‘samples’ can capture the underlying biology in the herds — it seems that only a quite small proportion of adult females in each herd were collared (see Fig S4). Related to this, a small sample for a herd could largely underestimate the size of the space the full-size herd uses, and this can overestimate population density. In the analysis, calf mortality was used as a proxy for adult female caribou fitness, while the majority of calf mortality was due to predation; then, it remains unclear to what extent this ‘fitness’ outcome was explained by, or linked to, habitat selection and/or social behaviours of collared females as opposed to predation (i.e., how meaningful it is to relate this ‘fitness’ to the social and habitat selection behaviours of adult females given the fact that the majority of mortality is by predation). Besides, when estimating behavioural repeatability and plasticity, a brief explanation in the main text on how the BRNs framework works will be helpful for readers who are not familiar with the field or approach. And generally, my feedback would be that, it’s hard to track those sets of models and analyses in the subsection following line 207. Current it focused extensively on what have been done but there lacks explanations on why they are done and on how one analysis was linked to another. I’d therefore suggest to give more explanations for the models and analyses following 1) why they are used, 2) how they work in relation to one another, and 3) how data was applied. Another concern I’d raise is related to habitat selection. If I understood correctly, for most of the time each herd was geographically distributed differently, and this made me wonder to what extent the availability of resource/habitat type for a given herd would differ with that for another herd (we know that natural habitats are spatiotemporally heterogeneous). If they did experience significant difference in habitat type availability, then the one had a lower availability would seemingly more likely to be ‘specialists’ than generalists. Thus, I’d suggest the analyses also take habitat availability into consideration, which requires more widespread habitat classifications than just those around the GPS fixes.

Following are some line-specific comments which might be helpful.

Line 20: It's unclear how they are 'inhibited'.

Line 30: Here 'ability' is vague - what exactly does it mean?

Line 50-51: Why is this expected?

Line 52: Which are 'well-connected'?

Line 54: How to change?

Line 57: It's unclear what exactly is 'ignored'. I think it's important to dive into this to clarify how exactly the current study fits into the literature.

Line 69: What does 'leads to equal fitness across unequal densities' mean?

Line 70: Some explanations on how it is extended will be helpful. Current readers are left wondering how the predictions for this study were proposed based on these.

Line 70-72: Here it's unclear on how density is linked to specialization - what are the mechanisms? How is resource quality involved? What are the underlying assumptions?

Line 98: What is 'social habitat specialization behaviour'?

Line 99: It's unclear what exactly the challenge is, and how does this study address the 'challenge'? By means of GPS tracking?

Line 113: While the same logic may apply, I'd say caribous are not gases but social mammals.

Line 120: It's unclear how it leads to the statement 'therefore predict no repeatability'. How does repeatability *per se* linked to IFD?

Line 121-122: It's unclear how this prediction (P4b) was made - what exactly is the 'behavioural ecology theory'?

Line 124-125: How is this predicted?

Line 127: How exactly is this prediction (P6a) 'based on the IFD'?

Line 130: What exactly is the 'ecological niche theory' in relevance?

Line 133: The table is basically repeating what were shown in the texts in the introduction. I'd suggest cut this and save space for clarifications on how those predictions were made based on the discussions in the literature.

Line 139: What about the males? Why only females?

Line 158: How were the surveys done? As an independent study, a brief description is necessary rather than simply give a ref.

Line 163: Explain how the data was scaled and centred, and why convergence was needed.

Line 168-169: Given that not all of herds overlapped, why they were put into a single network?

Line 171-174: It's unclear on why these particular spatio-temporal thresholds were used - any evidence to justify this?

Line 178-179: These lines did not really tell why the current approach was chosen. What exactly are the arguments in the literature, and what's the relevance to this study?

Line 187: Give citations for the index — probably move Bolnick et al. 2002 to the main text and briefly explain how the index works.

Line 199: The supplementary figure referred to was wrong.

Line 202: Given that a 'step' takes 2h (line 145), how reliable are the estimates of the calf mortality — is there any quantitative assessment? Instead of just giving ref 56, adding some more details on how calf mortality was inferred will be helpful to convince readers the reliability of the fitness estimates.

Line 217-219: The flow here is unclear. Line 217 starting with 'Although' just repeats what was said before it (and it has already been claimed that multi-variate models were used to avoid the issue), and it still remains unclear on how the issue was 'limited': how was the assessment done, and how does it work to address the issue? Some more in-depth mechanistic explanations are needed.

Line 301: As this line says, the black line in the 2nd panel of Fig 3 should be removed.

Line 356: How does 'generalizable' guarantee robustness?

Line 387: As mentioned earlier, it's unclear how much the 'fitness' used in this study was explained by the behaviours vs predation. Then, how could the study really 'delineate' the effects?

Reviewer #2 (Remarks to the Author):

This is a nice contribution to science that would benefit from a few revisions. I found the analysis difficult to follow and would benefit from an equation describing the statistical model. Multiple dependent variables is hard for me to conceptualize but perhaps it would make more sense if I could see the model explicitly. I have little use for Bayesian methods, especially when tied to uninformative priors. Subhash Lele has a paper that shows how these Bayesian excursions can lock into solutions that are not stable.

Conceptually I was puzzled that we have 2 choices: IDP and NVH. What about the ideal despotic distribution? Is that not an option for caribou? I came up with the idea for the 2 McLoughlin et al papers where the idea was to contrast a highly territorial roe deer with red deer. But the social dynamics of red deer turn out to be very similar to what we saw for roe deer albeit subtle hierarchies that are not as apparent. Something seems missing here given that Fretwell and Lucas offered the IDD as an alternative to the IFD.

A few minor details need to be fixed. Coyotes are *Canis latrans*, not *C. lupus*. The species that we studied in ref 67 was roe deer, not red deer.

Overall a very nice paper.

Mark Boyce, Alberta

REVIEWER COMMENTS

Reviewer #1 (Remarks to the Author):

Comments on manuscript NCOMMS-23-00651-T

The study presented in the manuscript tested the predictions generated based on the ideal free distribution (IFD) and the niche variation hypothesis (NVH) on how habitat specialization, social association strength and calf survival of 6 female caribou (*Rangifer tarandus*) herds covary under different spatiotemporal population densities. It has been shown that both habitat specialization and social association strength increase with population density, while habitat specialists are less social than generalists, thus supporting the NVH which posits that individuals should be more specialized as population density increases. Besides, it has been found that habitat specialization or generalization had no effect on fitness at low population density. The manuscript was well written and is novel to read — it tries to bring the IFD, the NVH, social behaviours, and their fitness consequences together to generate new insights. After reading, I'd like to feedback on a few things that might be worthwhile to think about.

Response: we wish to thank the reviewer for the helpful comments on our manuscript. See detailed comments to each concern below.

The first concern I'd raise relates to the descriptions of the predictions by the hypotheses. In particular, the manuscript lacks mechanistic explanations on what led to the different predictions by the NVH and the IFD following the state-of-the art discussions on them, and how the predictions specifically in this study were made based on a thorough review of these hypotheses. This also relates to the clarification on what the specific objectives the manuscript is aiming to achieve — what the study contributes to the literature in relation to the existing theory on IFD, NVH and social behaviour in particular. From my understanding, both the IFD and the NVH generate predictions on density and specializations, while the present study also incorporates the social aspects; however, the introduction did not clearly link these topics together with their own arguments, but keep generating predictions to test. I'd thus suggest more explanations on how these predictions were generated based on existing studies.

Response: we have re-written our predictions accordingly. See detailed comments below.

Unclear also remains in the method section. For example, it was unclear why only females were tracked and studied (and this left readers wondering how reliable the conclusions are when they are drawn from a completely female-biased dataset), and how reliable those collared adult females as 'samples' can capture the underlying biology in the herds — it seems that only a quite small proportion of adult females in each herd were collared (see Fig S4). Related to this, a small sample for a herd could largely underestimate the size of the space the full-size herd uses, and this can overestimate population density. In the analysis, calf mortality was used as a proxy for adult female caribou fitness, while the majority of calf mortality was due to predation; then, it remains unclear to what extent this 'fitness' outcome was explained by, or linked to, habitat selection and/or social behaviours of collared females as opposed to predation (i.e., how meaningful it is to relate this 'fitness' to the social and habitat selection behaviours of adult females given the fact that the majority of mortality is by predation). Besides, when estimating behavioural repeatability and plasticity, a brief explanation in the main text on how the BRNs framework works will be helpful for readers who are not familiar with the field or approach. And generally, my feedback would be that, it's hard to track those sets of models and analyses in the subsection following line 207. Current it focused extensively on what have been done but there lacks explanations on why they are done and on how one analysis was linked to another. I'd therefore suggest

to give more explanations for the models and analyses following 1) why they are used, 2) how they work in relation to one another, and 3) how data was applied.

Response: we now provide context for each analysis at the start of each section in the methods. We indicate why it was used, how it worked, and what data were used. See start of each methods section.

Another concern I'd raise is related to habitat selection. If I understood correctly, for most of the time each herd was geographically distributed differently, and this made me wonder to what extent the availability of resource/habitat type for a given herd would differ with that for another herd (we know that natural habitats are spatiotemporally heterogeneous). If they did experience significant difference in habitat type availability, then the one had a lower availability would seemingly more likely to be 'specialists' than generalists. Thus, I'd suggest the analyses also take habitat availability into consideration, which requires more widespread habitat classifications than just those around the GPS fixes.

Response: indeed, the reviewer makes an excellent point here and we appreciate it very much! We have in fact already done this type of analysis and it is presented in Appendix S4 and Figure S7. We acknowledge that such an important idea is indeed deep in the supplement, and that the supplement alone is likely the topic of a very interesting paper. However, to keep our core narrative as streamlined as possible, we have elected to retain it in the supplement. In the supplement we provide extensive discussion about how we account for habitat availability and how habitat specialization is related to habitat selection. Specifically, in the caption of figure S7, we note that "For high fitness individuals, habitat specialists tend to have uneven selection. Specialists had stronger selection for a single habitat type and either had equivocal selection or avoided other habitat types. Note high-fitness specialists occupy more data space than low-fitness generalists. In contrast to high-fitness specialists, low-fitness generalists tended to be more even and more equivocal in their selection, neither select nor avoid any habitats."

Following are some line-specific comments which might be helpful.

Line 20: It's unclear how they are 'inhibited'.

Response: we have reworded to read as "were less social to avoid competition". Line 21.

Line 30: Here 'ability' is vague - what exactly does it mean?

Response: we have replaced "ability" with "how" to read "...how animals use space, interact with conspecifics, and copy their genes". Line 31.

Line 50-51: Why is this expected?

Response: at high population density competition for resources (food/mates) is implicitly expected to increase. This is why isoclines from IFD have negative slopes, because per-capita fitness declines with density (Morris 2003). We have rephrased to read "...changes in density to reduce the potential for competition associated with higher population density".

Line 52: Which are 'well-connected'?

Response: we have replaced “well-connected” with “highly social”. Line 54.

Line 54: How to change?

Response: we have added an example following the sentence in about how the relationship between sociality and fitness might change as a function of population density. The example reads as: “For example, if increasing density results in higher individual sociality, we might predict the most social individuals have higher fitness, however, if increasing density reduces individual sociality, we might predict the most social individuals have lower fitness (Albery et al. 2021).” line 58.

Line 57: It’s unclear what exactly is ‘ignored’. I think it’s important to dive into this to clarify how exactly the current study fits into the literature.

Response: we have added an additional sentence which highlights numerous examples linking sociality and density in the literature, but that few studies test how the relationship between sociality and fitness varies as a function of population density. The new sentence reads: “While studies highlighting the link between sociality and density have become increasingly common (Strickland et al. 2018; Strickland and Frere 2020; Albery et al. 2021), few empirical studies explicitly quantify individual sociality and fitness and assess how this relationship varies across a gradient of population density.” Lines 63-65.

Line 69: What does ‘leads to equal fitness across unequal densities’ mean?

Response: unequal density referred simply to different density. The theory is that under an IFD, mean fitness should be equal in each habitat despite each habitat having different carrying capacities. See lines 79-81.

Line 70: Some explanations on how it is extended will be helpful. Current readers are left wondering how the predictions for this study were proposed based on these.

Response: we have completely re-written this paragraph and provided further explanation about the link between IFD, density-dependent habitat selection and competition, which is the implied mechanism driving density-dependent habitat selection (see also Morris 2003, Morris 2011). See lines 84-88.

Line 70-72: Here it’s unclear on how density is linked to specialization - what are the mechanisms? How is resource quality involved? What are the underlying assumptions?

Response: At least since Rosenzweig and isoclines, population density has been thought to be correlated with competition. Thus the mechanism for density dependence is competition. We have expanded this section and added significant text to address questions of mechanisms, resource quality, and the assumptions.

“Habitat selection phenotypes vary among individuals²³, across densities²⁴, and infer foraging behaviour²⁰ --- where citation 20 is Rosenzweig (1981) – line 72-73.

Line 98: What is ‘social habitat specialization behaviour’?

Response: this was a typo and should read “social behaviour and habitat specialization”

Line 99: It's unclear what exactly the challenge is, and how does this study address the 'challenge'? By means of GPS tracking?

Response: indeed, we have revised accordingly. The sentence now reads "...remains a challenge given the difficulty in simultaneously measuring social and spatial behaviours⁴ and here we address the challenge by using GPS relocations to estimate these behaviours for individual animals."

Line 113: While the same logic may apply, I'd say caribous are not gases but social mammals.

Response: We agree that animals are not gas particles. Ideal Gas Law, however, has been a useful model for animal movement. We have changed the wording to read "We predicted that individual values of social strength should increase with population density based on the expectation that higher density increases the probability of interaction" (lines 152-153).

Line 120: It's unclear how it leads to the statement 'therefore predict no repeatability'. How does repeatability *per se* linked to IFD?

Response: the IFD predicts patterns of habitat selection at the population level: IFD is agnostic to individuals, as they were not the focus of study when the idea was generated. Whether individual A is in habitat A or B, or switches between habitat A or B, is not considered when measuring the IFD. We therefore have no a priori prediction.

Line 121-122: It's unclear how this prediction (P4b) was made - what exactly is the 'behavioural ecology theory'?

Response: we have deleted "behavioural ecology theory" and reworded to read "...while the logical extension of the NVH when considering the explicit effects of competition on niche variation we predict that.."

Line 124-125: How is this predicted?

Response: we have revised the predictions - see line 160.

Line 127: How exactly is this prediction (P6a) 'based on the IFD'?

Response: we have removed "based on the IFD".

Line 130: What exactly is the 'ecological niche theory' in relevance?

Response: we have deleted "ecological niche theory".

Line 133: The table is basically repeating what were shown in the texts in the introduction. I'd suggest cut this and save space for clarifications on how those predictions were made based on the discussions in the literature.

Response: we have moved the table to the supplementary materials (see Table S1)

Line 139: What about the males? Why only females?

Response: as stated at line X, our GPS collar data were collected by the Newfoundland and Labrador Wildlife Division. In ungulates, adult females are considered the most important demographic class (Gaillard and Festa-Bianchet 1998, 2000) for population dynamics. Furthermore, where adult females have little variance (elasticity) in their survival rates, offspring survival – or annual reproductive success – is highly variable. Thus, because variance in offspring survival it is an important and promising demographic class from which to detect effects. For these reasons, most management agencies (including the NLWD) deploy collars on adult females as the objectives of these agencies is to answer questions about vital rates (e.g. reproductive success) and to monitor survival of calves. Given these management objectives, in combination with the cost of deployment and maintenance of collars, adult males are not typically collared by management agencies.

Line 158: How were the surveys done? As an independent study, a brief description is necessary rather than simply give a ref.

Response: we have explained how surveys were conducted and we now cite Ellington et al. 2020 who provide extensive background on the surveys conducted in our study system.

Line 163: Explain how the data was scaled and centred, and why convergence was needed.

Response: as noted, data were scaled and centred to have a mean of zero. The reason for this is because the actual values of social strength and Psi are on different scales, and to facilitate model convergence, it is appropriate to scale variables. Alternatively, should a model *not* converge, it effectively stops “running”, i.e., there are no results. This is a common issue when variables are on different scales (Carlin and Cowles 1996). We have provided a similar explanation at line X of the revised manuscript.

Line 168-169: Given that not all of herds overlapped, why they were put into a single network?

Response: there is inter-change of individuals among herds (and by binning individuals into herds, we removed some associations).

Line 171-174: It's unclear on why these particular spatio-temporal thresholds were used - any evidence to justify this?

Response: collars were programmed to record locations every 2 hours (e.g. 6am, 8am, 10am, etc). However, it is rare for a fix time to be exactly “on the hour”, i.e. at 6:00am. In most cases there is some minor deviation (usually only a few seconds, but sometimes up to a few minutes), which simply requires that we incorporate a small measure of temporal error (i.e. 5 minutes) around each (bi)hourly fix. The 50 m spatial threshold is taken from Kasozi and Montgomery (2020), who identify 50 m as a standard grouping threshold when observing ungulate groups.

Line 178-179: These lines did not really tell why the current approach was chosen. What exactly are the arguments in the literature, and what's the relevance to this study?

Response: we provide some further context, but the debate within the social network literature is extensive and complicated. Arguably, much of this is beyond the scope of our article and we provide basic information for readers to be aware of the issue. In the revised manuscript at line 228-237, we note:

“Given recent discussion regarding the use of effect sizes and Bayesian inference to model social networks⁶², we did not generate null models and estimate effects of covariates on social network strength in a multi-variate regression framework. Specifically, the social network null modelling framework, proposes data-stream permutations should be conducted to ensure observed measures of sociality are non-random. In many cases, common data stream permutations are not appropriate for hypothesis testing when using regression models (Weiss et al. 2020). Specifically, data stream permutations can result in high type I error rates and the null expectation of a random network is not appropriate in most systems (Weiss et al. 2020). Moreover, when using Bayesian regression models, the traditional data stream to model coefficient pipeline (Weiss et al. 2020; Farine 2017) is not appropriate due to the iterative nature of Bayesian statistics (Hart et al. 2022). However, despite concerns with social network permutation techniques, there remains an expectation to ensure observed relationships are indeed non-random.”

Line 187: Give citations for the index — probably move Bolnick et al. 2002 to the main text and briefly explain how the index works.

Response: fixed as suggested. See line 246-251.

Line 199: The supplementary figure referred to was wrong.

Response: we have now moved this appendix to the main text, so no reference to the appendix required here.

Line 202: Given that a ‘step’ takes 2h (line 145), how reliable are the estimates of the calf mortality — is there any quantitative assessment? Instead of just giving ref 56, adding some more details on how calf mortality was inferred will be helpful to convince readers the reliability of the fitness estimates.

Response: we previously provide these details in Appendix S2, but we have moved all of Appendix S2 (~100 words) to the main text for further clarity and to respond to this comment. See lines 264-271.

Line 217-219: The flow here is unclear. Line 217 starting with ‘Although’ just repeats what was said before it (and it has already been claimed that multi-variate models were used to avoid the issue), and it still remains unclear on how the issue was ‘limited’: how was the assessment done, and how does it work to address the issue? Some more in-depth mechanistic explanations are need.

Response: we provide further mechanistic explanation at lines 282-297. In fact, bivariate models remove (not limit) the confound associated with stats on stats as there are no stats-on-stats happening (i.e. an output/coefficient from one model being used in a second model).

Line 301: As this line says, the black line in the 2nd panel of Fig 3 should be removed.

Response: fixed as suggested.

Line 356: How does ‘generalizable’ guarantee robustness?

Response: we have removed this sentence.

Line 387: As mentioned earlier, it's unclear how much the 'fitness' used in this study was explained by the behaviours vs predation. Then, how could the study really 'delineate' the effects?

Response: we have changed the word "delineate" to "test". See line 460.

Reviewer #2 (Remarks to the Author):

This is a nice contribution to science that would benefit from a few revisions. I found the analysis difficult to follow and would benefit from an equation describing the statistical model. Multiple dependent variables is hard for me to conceptualize but perhaps it would make more sense if I could see the model explicitly. I have little use for Bayesian methods, especially when tied to uninformative priors. Subhash Lele has a paper that shows how these Bayesian excursions can lock into solutions that are not stable.

Response: thank you for your generous comments and for providing comments in the word document as well.

We have added the model equation in the appendix (line 294) and we now cite Lele 2020.

Conceptually I was puzzled that we have 2 choices: IDP and NVH. What about the ideal despotic distribution? Is that not an option for caribou? I came up with the idea for the 2 McLoughlin et al papers where the idea was to contrast a highly territorial roe deer with red deer. But the social dynamics of red deer turn out to be very similar to what we saw for roe deer albeit subtle hierarchies that are not as apparent. Something seems missing here given that Fretwell and Lucas offered the IDD as an alternative to the IFD.

Response: this is a reasonable comment given the inherent links between IFD and IDD. We now introduce IDD early in the introduction (line 82) and we follow up later in the introduction to explicitly state that we do not test the IDD for caribou (line 151-152). Specifically, the IDD effectively states that habitat selection is governed by dominant animals suppressing the space use of subordinates. Caribou are not territorial per se (as are roe deer), though we do think the IDD may apply to fine-scale caribou foraging during winter at craters. Dominants may suppress space use of subordinates at the local/patch scale to gain access to (and defend) higher quality cratering sites. However, the spatial and temporal scale of the data we used for our analyses (i.e. habitat classes across seasons) likely inhibit us from developing an appropriate test of the IDD, particularly within this already complicated contribution.

A few minor details need to be fixed. Coyotes are *Canis latrans*, not *C. lupus*. The species that we studied in ref 67 was roe deer, not red deer.

Response: fixed as suggested.

Overall a very nice paper.

Response: thank you again for your generous comments on our manuscript.

Mark Boyce, Alberta

REVIEWER COMMENTS

Reviewer #1 (Remarks to the Author):

Please see in the pdf attached.

REVIEW COMMENTS ON MANUSCRIPT NCOMMS-23-00651A

General comments:

The authors have made extensive progress improving the strength and clarity of the manuscript. After re-reading I'd raise two main concerns along with some specific comments which may help. First, in the introduction unclarity still remains in relation to how those predictions to test were laid out based on the two fundamental predictions by the IFD and NVH. Related to this, given that this study focuses on testing the IFD and NVH and that these hypotheses have their own underlying assumptions, it's also critical to clarify how the system, characterized by a strong social component, would work for testing both and generating new insights in relation to sociality (e.g., should these predictions be adapted for the social caribou, and if so, how the predictions would then be and why (e.g., see specific comments for L84, L100, L155-174). The other concern I'd raise is on the use of offspring survival as a measure of adult fitness in relation to their resource use/selection (e.g., see specific comments for L59 below).

Specific comments — main texts:

L20: Isn't it that specialists tend to be less social than generalists according to Fig 2? How is this concluded the other way around?

L21: The previous line just said that specialists are 'less social' than generalists, but here it is concluded the generalists are. Besides, can it be more conclusive than just saying 'suggesting the possibility that generalists were less social to avoid competition'?

L30: 'but as population density increases, resources become limited' — they are always limited (i.e., non-infinity) no matter whatever density.

L42-50: Why are eco-evolutionary dynamics introduced here? Seriously, how relevant are they to the present study? What eco-evolutionary insights can these results generate?

L52: 'reduce' — this may not always be the case (e.g., consider what dominant individuals may do).

L54: 'connections of an individual to other highly social individuals' — this may not be the definition of social network centrality. I guess what it meant was the extent to which an individual is socially connected with others (e.g., node degree in a social network).

L56: How to change?

L58: 'higher fitness' — why? Higher sociality is not always associated with higher fitness. Increases in density may not always led to higher competition — e.g., when resource carrying capacity is high relative to a given density.

L59: 'lower' — why? This has to be spell out clearly. Being more social isn't always a bad thing - imaging if predation pressure is increased, being in a larger group may be less likely to be eaten (i.e., sociality is typically driven by trade-offs). I'd suggest to be specific about the contexts — in this study it's mostly in a foraging context. This reminds me of my earlier comment on the accounting of density-dependence on offspring survival: given the narrative is mostly framed under density-dependent foraging context of adults, to what extent resource availability vs predation explains offspring survival rate? The authors pointed out that the majority of calf mortality was due to predation, and I'm still wondering, given this fact, how meaningful is it to use calf survival as a proxy of adult fitness in relation to resource use or foraging — especially offspring have an unweaned life history stage (and offspring mortality inferred by GPS data is mostly likely in this stage — L265-272). Justifying the use of this proxy is important.

L63: 'ignored' — not all the studies are dedicated to evaluate the adaptive value of social behaviours. Spell out what the consequences of being 'often ignored' are.

L64-65: This statement is still unclear on what gaps this study is trying to address. To come up with a stronger one, more thorough literature review is needed (e.g., Silk, J. B. 2007. The adaptive value of sociality in mammalian groups. *Philos. Trans. R. Soc. B*, 362(1480), 539-559).

L74: Introduction to the theories comes too late — given that the whole manuscript is about testing competing hypotheses by these theories (as it is claimed in the abstract). See also comments for L155-174.

L82: In the foraging context, if there's solid evidence suggesting IDD is not applicable to the species, then it may be omitted — if the other reviewer also agrees.

L84: Give evidence/facts to show how the study system/species meets the underlying assumption. Besides, in such a social species, how would sociality violate the assumption of habitat selection underlying IFD?

L89: Again, I'd expect this applies when density is above a threshold under a given carrying capacity. What is 'the scale of social interactions'?

L93: 'therefore sociality' — isn't sociality itself a direct component underlying IFD?

L97: How to reduce? Who, generalists vs specialists, will give up their current patch if they are to 'reduce' competition? If there's no dominance, why would they be willing to leave? What incentives could be?

L100: How does sociality (or social forces) of the species violate this 'ideal free scenario'? This relates to the testability of the IFD by the social species in focus, and this need to be justified.

L102: 'reduce' — be clearer on how this is deduced.

L105-106: There's a bit confusion — why does resource quality come into play here (while in the example given it's just about density)? Why not just focusing on discussing the effects of density? See also comments for L97 and L100.

L139: What does 'appear to' imply? See also comments for L100.

L152: 'non-territorial and typically do not defend resources' — then why IDD is mentioned given that it's irrelevant/inapplicable in this case? Give citation(s)/evidence to support this.

L155-174: While I got the two fundamental predictions by the IFD (P2a) and the NVH (P2b), as an ordinary reader I'd say I'm still struggling to get my head around how the subsequent predictions (P4b, P5a, P6a, P6b) were laid out. As my earlier comment, I'd suggest to be more clear on the mechanisms by which these were made. This is central to the manuscript and failure in making these logic links clear does not allow the study to hone in the existing theory in the literature. At the current stage, sorry I cannot understand the logic behind.

L159: 'more social' — there's a bit of confusion in this term. From such spatial association data it's not easy to tell which individual are actually more social. Imagine an extreme social individual exists in a small herd. Its social network centrality may consistent be lower than an actually less social individual in a much larger herd, simply because the latter is more likely to be observed next to more of others in the larger herd (i.e., the availability of immediate social mates). Related to this, it may be more informative if centrality is scaled with herd size.

L180: 'female' — while females have been collared for conservation management purposes, this study did try to use the dataset to test different questions. I'd suggest to acknowledge this fact and use a few lines to discuss the potential consequences such a sex-biased dataset may or may not have for the conclusions (e.g., to what extent do the authors believe these conclusions also apply to the

males or both sexes?).

L654: 'maximize' — not always I'd say.

L659: 'greatest' — the wording is rather extreme!

L667: 'therefore' — this may not be deduced — what if the effects sum up to zero across the whole population?

Specific comments — supplement:

L73: Does this number apply to all networks? Does this need to be adjusted with network size?

L76: Remove 'and social strength'?

L77: 'non-random' — does this mean social strength is not linked to offspring survival? Unclarity remains in the interpretations of this figure.

L121: Variance should be relatively high.

L141: This info seems to be the other way around.

L145: 'specialists' — but this is conditioned on high vs low fitness.

L206: It would be more informative if the measures of variance are labelled. It seems only a few individuals exhibited large variation in the left panel.

REVIEW COMMENTS ON MANUSCRIPT NCOMMS-23-00651A

General comments:

The authors have made extensive progress improving the strength and clarity of the manuscript. After re-reading I'd raise two main concerns along with some specific comments which may help. First, in the introduction unclarity still remains in relation to how those predictions to test were laid out based on the two fundamental predictions by the IFD and NVH. Related to this, given that this study focuses on testing the IFD and NVH and that these hypotheses have their own underlying assumptions, it's also critical to clarify how the system, characterized by a strong social component, would work for testing both and generating new insights in relation to sociality (e.g., should these predictions be adapted for the social caribou, and if so, how the predictions would then be and why (e.g., see specific comments for L84, L100, L155-174). The other concern I'd raise is on the use of offspring survival as a measure of adult fitness in relation to their resource use/selection (e.g., see specific comments for L59 below).

Response: we thank the reviewer for their helpful comments on our manuscript. Specifically, we wish to re-iterate our over-arching response to each of the two main concerns listed here (See also cover letter).

- 1) **Assumptions of IFD in a social system (and predictions).** As requested by the editor, we have also make specific note of how the social caribou system can be adapted to test these predictions at lines 141-151 of the introduction. Given the remaining unclarity regarding some of our predictions, we have edited our predictions down and rephrased some of the previous predictions as objectives (see lines 178-190 and Table S1).
- 2) **Use of offspring survival as a measure of adult fitness.** We now place greater emphasize on the population biology of caribou, and ungulates in general and the use of annual reproductive success (calf survival) as a proxy for fitness. Ungulates (including caribou) tend to have low variance in adult survival, but high variance in annual reproductive success due to high mortality of neonates. We are therefore confident in our use annual reproductive success as a proxy for fitness and we justify this at lines 138-140 of the manuscript and lines 12-17 of the appendix.

See below for detailed responses in blue font to each comment.

Specific comments — main texts:

L20: Isn't it that specialists tend to be less social than generalists according to Fig 2? How is this concluded the other way around?

Response: this was an error, we have fixed it accordingly to state that specialists are less social than generalists. See line 20.

L21: The previous line just said that specialists are ‘less social’ than generalists, but here it is concluded the generalists are. Besides, can it be more conclusive than just saying ‘suggesting the possibility that generalists were less social to avoid competition’?

Response: this was an error, we have fixed it accordingly to state that specialists are less social than generalists See line 21.

L30: ‘but as population density increases, resources become limited’ — they are always limited (i.e., non-infinity) no matter whatever density.

Response: we have reworded to read “as population density increases, resources become increasingly limited”. Density-dependent habitat selection (IFD and NSH) all make clear that as population density increases per capita resources are reduced. Typically, this results in density dependence in survival and reproduction, where when per capita resources are higher so too is survival and reproduction, leading to population increase, for example. – see lines 30-31. Thus, it stands to reason that when populations are growing there are more available resources per capita. The inverse then is true when density exceeds the available resources, or the carrying capacity of the environment. Here we anticipate population declines because of decreased survival. In ungulates, the first place to detect population declines due to resource limitation is decreased offspring survival¹⁻³. We now are clearer about the links between population size and resource availability at lines 138-140.

L42-50: Why are eco-evolutionary dynamics introduced here? Seriously, how relevant are they to the present study? What eco-evolutionary insights can these results generate?

Response: we have removed mention of eco-evolutionary dynamics in the introduction. The new text in this section simply references to density-dependence (see lines 42-46).

L52: ‘reduce’ — this may not always be the case (e.g., consider what dominant individuals may do).

Response: we have revised the sentence to read:

“Density fluctuates in natural populations, suggesting that individuals should display behavioural plasticity in response to fine-scale spatiotemporal changes in population density”. See lines 46-48.

L54: ‘connections of an individual to other highly social individuals’ — this may not be the definition of social network centrality. I guess what it meant was the extent to which an individual is socially connected with others (e.g., node degree in a social network).

Response: we have revised to read:

“For gregarious species, social network centrality (i.e., the extent to which an individual is socially connected to others)...”. See lines 49-50.

L56: How to change?

Response: fixed.

L58: ‘higher fitness’ — why? Higher sociality is not always associated with higher fitness. Increases in density may not always led to higher competition — e.g., when resource carrying capacity is high relative to a given density.

Response: Please see comment above re per capita resources, competition, and population dynamics. We note that one of our key findings is that sociality does not improve reproductive success, but that the correlation between sociality and habitat specialization could have misattributed social affects in the absence of quantifying the habitat selection behaviors. The vast majority of work demonstrates that more social individuals tend to have higher fitness ^{4,5} (at least for gregarious species). We have chosen not to change the wording here.

L59: ‘lower’ — why? This has to be spell out clearly. Being more social isn’t always a bad thing - imaging if predation pressure is increased, being in a larger group may be less likely to be eaten (i.e., sociality is typically driven by trade-offs).

Response: Our measure of density is at the population level, not the group. Based on the expectations of density-dependent habitat selection and density dependence in population dynamics, higher density equates to higher competition and the prediction of lower fitness for social individuals. Indeed, other predictions may be generated when considering other ecological processes, but to ensure the narrative remains simple and sets up our study, we have chosen not to introduce the predator dynamic to our introduction. Indeed, at fine scales there may be local dynamics that mitigate risk. In general, however, habitat selection accounts for local/group density, variation in vigilance in groups.

I’d suggest to be specific about the contexts—in this study it’s mostly in a foraging context. This reminds me of my earlier comment on the accounting of density-dependence on offspring survival: given the narrative is mostly framed under density-dependent foraging context of adults, to what extent resource availability vs predation explains offspring survival rate?

Response: We want to take a moment to clarify that the study is more precisely about density-dependent habitat selection, which is not exactly the same as density-dependent foraging. Habitat selection implicitly accounts for the risks of acquiring resources and is grounded in the key assumption that animals select habitats that provide equal fitness benefits given the density. Our analysis partitions resource types (see methods 413-415 and lines 114-134 of the appendix) as a response to density.

The authors pointed out that the majority of calf mortality was due to predation, and I’m still wondering, given this fact, how meaningful is it to use calf survival as a proxy of adult fitness in relation to resource use or foraging — especially offspring have an unweaned life history stage (and offspring mortality inferred by GPS data is mostly likely in this stage — L265-272). Justifying the use of this proxy is important.

Response: Once caribou recruit into the population (i.e., are sub adults and adults), there is very little variance in survival, particularly adult female survival. However, there is quite a bit of variance in calf survival¹⁻³. During the period of interest calves depend entirely on their mothers. Thus, our paper is framed from the perspective of the mother's fitness. – or the mother's reproductive success. We now frame our idea using the word 'fitness', but when we refer to the results and their interpretation we use 'annual reproductive success', as that is what we measured. For example, it doesn't matter how social or how much of a specialist an adult is because their probability of survival in a given year is upwards of 90%. We now make reference to this notion at lines 138-140 of the introduction and lines 12-17 in the appendix.

L63: 'ignored' — not all the studies are dedicated to evaluate the adaptive value of social behaviours. Spell out what the consequences of being 'often ignored' are.

Response: we have added an additional sentence: "As a result, there are few empirical examples that demonstrate the affect of population density on the sociality-fitness relationship." See lines 61-62.

L64-65: This statement is still unclear on what gaps this study is trying to address. To come up with a stronger one, more thorough literature review is needed (e.g., Silk, J. B. 2007. The adaptive value of sociality in mammalian groups. *Philos. Trans. R. Soc. B*, 362(1480), 539-559).

Response: we now cite Silk 2007 at line 61. While Silk is the definitive review linking sociality and fitness. There are, however, few mentions of how population density affects the relationship between sociality and fitness. Social behaviours have to occur in the context of a particular density and per-capita resources. We now try to be clearer that density driven social behavior or habitat specialization are the competing (or synergistic) pathways that could explain variance in offspring survival. The pathway – and this is likely true, but unarticulated – for many examples in Silk 2007 is: density → social behavior → fitness; not just social behavior → fitness. What is the causes changes in social behavior? Density and demography, likely. That the density pathway has a fitness (survival and reproduction) outcome that then changes the population density, further cements the importance of illustrating the potential for developing evidence for an eco-evolutionary pathway. We are now clearer about this at lines 56-62 in the introduction.

L74: Introduction to the theories comes too late—given that the whole manuscript is about testing competing hypotheses by these theories (as it is claimed in the abstract). See also comments for L155-174.

Response: we argue that setting up the density dependent literature first is important and is required context ahead of the IFD and NVH.

L82: In the foraging context, if there's solid evidence suggesting IDD is not applicable to the species, then it may be omitted — if the other reviewer also agrees.

Response: We agree as at the scales we are making inferences are more likely to have support for IFD. However, we do see merit in acknowledging that IDD relates to other systems that may provide different hypotheses and outcomes with regard to specialization and social behavior.

Thus, the inclusion of IDD was in response to the other reviewer and as a result we are hesitant to remove it from the introduction.

L84: Give evidence/facts to show how the study system/species meets the underlying assumption. Besides, in such a social species, how would sociality violate the assumption of habitat selection underlying IFD?

Response: The IFD has a small set of assumptions, e.g., cost-free movement, perfect knowledge of the habitat and fitness landscapes, and the ability to assort to maximize fitness. Sociality could violate some of those assumptions, but in some cases, these violations are what sparks new and exciting scientific discovery. We have made note of this in the manuscript at line 141-151.

L89: Again, I'd expect this applies when density is above a threshold under a given carrying capacity. What is 'the scale of social interactions'?

Response: What we meant was that competition often occurs in the same geographic space as social interactions and that these occur at a fine scale. We have added this to the sentence at line 86-87.

L93: 'therefore sociality'—isn't sociality itself a direct component underlying IFD?

Response: Sociality is not a direct underlying component of IFD. We agree, however, that the sentence is misleading or erroneous as competition is well studied in IFD. Rather we have rephrased the sentence to read: "Given density-dependent effects on competition likely influence social behavior we endeavor to disentangle the relative effects of social behavior and habitat selection." – see lines 90-92.

L97: How to reduce? Who, generalists vs specialists, will give up their current patch if they are to 'reduce' competition? If there's no dominance, why would they be willing to leave? What incentives could be?

Response: We have removed the word 'reduce' as it caused confusion. If two animals occupy the same patch (irrespective of dominance) they then have fewer per-capita resources between them. Thus, by not occupying the same patch as a conspecific, i.e., spacing away, one could have reduced competition by increasing the per capita resources immediately available. We have clarified this basic idea at line 94-96 "For example, when a social group spreads out in space, animals may not be feeding at the same fine-scale patch, thus increasing the per capita resource immediately available to each individual. Spacing out may also facilitate access to multiple types of patch or food."

L100: How does sociality (or social forces) of the species violate this 'ideal free scenario'? This relates to the testability of the IFD by the social species in focus, and this need to be justified.

Response: Fantastic question and indeed it highlights a key take home message of our paper. We are pleased at this point in the introduction we have piqued the reviewers interest in the key part of our narrative.

It is best to view the IFD as a null model. Its assumptions of perfect knowledge of habitat and fitness landscapes and cost-free movement are not bugs, but features that have long contributed to important tests of the model. Sociality, for example, could influence the IFD because social animals are not necessarily free to move among habitats. Indeed, that we identify a correlation between social connectedness and one's ability to specialize translates whereby one's ability to specialize positively affects fitness is one of our keystone, novel, and impactful findings.

We have made changes reflecting this sentiment:

“Caribou are social ungulates that live in fission-fusion societies⁴⁷ and at broader scales conform to the ideal free distribution (e.g., during calving⁴⁸). We view the IFD as a null model when considering animal space use. Assumptions of the IFD include cost free movement between patches and perfect knowledge of habitat²⁶. While these assumptions may be violated in systems with social animals, these violations mark important contributions to the field of behavioural ecology. For example, among relatives there is no free choice, suggesting that individuals maximizing inclusive fitness may overexploit habitats at a given density, even though classic IFD theory suggests they should re-assort to maximize fitness⁴⁹. Moreover, social animals are not necessarily free to move among habitats in an ideal way as the benefits of social foraging are predicted to outweigh the costs⁵⁰. These assumptions of the IFD have historically yielded novel insight into how space use of grouping animals reinforces social behaviours.” – see lines 141-151.

L102: ‘reduce’ — be clearer on how this is deduced.

Response: We now ensure that we cite Fretwell and Lucas⁷ at this point, as this idea is at the center of IFD: as density (competition) in habitat A increases, at some threshold animals will be able to have equal fitness gains by moving to habitat B. See lines 100-101.

It was not our intent in the paper to provide a primer on IFD. However, if the Associate Editor and Reviewer feel that readers will not have ready access to learn about IFD as a framework, its assumptions, and its rules, then we would be open to inserting a box into the paper called: “A primer on IFD”. But at the moment, it is not clear to us that this publication is the most appropriate for a pedagogical approach like a primer. We love the idea of IFD and if needed we can provide more general primer for readers.

L105-106: There's a bit confusion — why does resource quality come into play here (while in the example given it's just about density)? Why not just focusing on discussing the effects of density? See also comments for L97 and L100.

Response: We have tried to help readers who are unfamiliar with IFD and its predictions by following this statement with an empirical example. In IFD “quality” is a difficult concept – indeed it is for all habitat selection. High quality resources, as is the case for red deer, are found in habitats that would be occupied by animals when density is low. When density in those habitats increases the per capita gains from those resources decrease to the point where other

habitats will be occupied without compromising fitness. To help clarify we have rephrased to indicate that grasslands are considered high quality habitats for the red deer. See lines 106-107.

L139: What does 'appear to' imply? See also comments for L100.

Response: we have deleted the word “appear to” to reduce ambiguity. The word ‘appear’ was a careful self-reflection of our paper that was not an experimental test of IFD, but one based on observed patterns in nature. We are fine removing ‘appear’ to reduce ambiguity, but do want the reviewer and editors to know that it is our practice to be careful and conservative with our science and its interpretations.

L152: ‘non-territorial and typically do not defend resources’ — then why IDD is mentioned given that it’s irrelevant/inapplicable in this case? Give citation(s)/evidence to support this.

Response: again, IDD was included as per the suggestion of the previous reviewer 2. We felt their comment merited consideration and including IDD in the introduction felt appropriate.

L155-174: While I got the two fundamental predictions by the IFD (P2a) and the NVH (P2b), as an ordinary reader I’d say I’m still struggling to get my head around how the subsequent predictions (P4b, P5a, P6a, P6b) were laid out. As my earlier comment, I’d suggest to be more clear on the mechanisms by which these were made. This is central to the manuscript and failure in making these logic links clear does not allow the study to hone in the existing theory in the literature. At the current stage, sorry I cannot understand the logic behind.

Response: after some reflection, we have decided to alter how we pitch predictions 4-6. We agree with the reviewer that they were ill-conceived, and our initial attempt was to try and fit these objectives in as predictions. To address this comment, we now present the previous predictions 4 and 5 as objectives (now objectives 1 and 2, see Table S1) and have rewritten the text as follows:

“In addition to the predictions associated with IFD and NVH (see above), we also sought to quantify repeatability (r) of social strength and habitat specialization. Behavioural traits are typically considered highly repeatable if $r > 0.40$, moderately repeatable if $0.20 > r < 0.40$, and low or negligible repeatability of $r < 0.20$. Notably, our objective was to estimate repeatability to place social strength and habitat specialization within a broader behavioural and evolutionary ecology context (objective 1 listed in Table S1). Finally, the IFD does not explicitly incorporate inter-individual behavioural variation. Our objective was to explore the relationship between annual reproductive success and sociality across density through the lens of the IFD. We might expect that at lower density, annual reproductive success would be highest for more socially connected individuals due to reduced competition at low density, while at higher density (and therefore higher competition) annual reproductive success would be highest for less social individuals that seek to avoid competition (Objective 2). For details on all predictions and objectives see Table S1.” – see lines 178-190.

L159: ‘more social’ — there’s a bit of confusion in this term. From such spatial association data it’s not easy to tell which individual are actually more social. Imagine an extreme social

individual exists in a small herd. Its social network centrality may consistently be lower than an actually less social individual in a much larger herd, simply because the latter is more likely to be observed next to more of others in the larger herd (i.e., the availability of immediate social mates). Related to this, it may be more informative if centrality is scaled with herd size.

Response: changed to “more socially connected”, which is a more appropriate network term and reflects better what we measured. See line 169-170.

L180: ‘female’ — while females have been collared for conservation management purposes, this study did try to use the dataset to test different questions. I’d suggest to acknowledge this fact and use a few lines to discuss the potential consequences such a sex-biased dataset may or may not have for the conclusions (e.g., to what extent do the authors believe these conclusions also apply to the males or both sexes?).

Response: we have included the following in the appendix:

Collars were deployed on adult female caribou for one to three years, but collars were often re-deployed on the same individuals for up to seven years. Consistent with most ungulate monitoring practices in North America, including elk⁸ and caribou⁹, only adult female caribou were monitored using GPS collars during the period of our study. In ungulates, adult females are considered the most important demographic class^{1,10} for population dynamics. Furthermore, where adult females have little variance (elasticity) in their survival rates, offspring survival – or annual reproductive success – is highly variable. Thus, because variance in offspring survival it is an important and promising demographic class from which to detect effects. For these reasons, most management agencies (including the Newfoundland and Labrador Wildlife Division) deploy collars on adult females as the objectives of these agencies is to answer questions about vital rates (e.g. reproductive success) and to monitor survival of calves. Given these management objectives, in combination with the cost of deployment and maintenance of collars, adult males are not typically collared by management agencies. See lines 8-21 of the appendix.

L654: ‘maximize’ — not always I’d say.

Response: edited to read “Animals use space, select habitat, and occupy social positions that are intended to maximize their fitness.” – see lines 320-321.

L659: ‘greatest’ — the wording is rather extreme!

Response: replaced with high to read “We present evidence supporting predictions of the NVH that highlight the adaptive value of individual habitat specialization was high at high population density” – See line 326.

L667: ‘therefore’ — this may not be deduced — what if the effects sum up to zero across the whole population?

Response: we have removed the word “therefore”.

Specific comments — supplement:

L73: Does this number apply to all networks? Does this need to be adjusted with network size?

Response: yes, the 50m buffer applies to all networks. We do not adjust the 50m based on network size as this would create asymmetry in the networks due to some having higher spatial thresholds for grouping (and therefore presumably would be more socially connected).

L76: Remove ‘and social strength’?

Response: fixed.

L77: ‘non-random’ — does this mean social strength is not linked to offspring survival? Unclarity remains in the interpretations of this figure.

Response: non-random simply means that the observed coefficient fell outside the distribution of randomly generated coefficients. We have changed the text to read: “we also developed data-stream permutations to assess the potential for non-random (i.e. social associations that arise by chance as opposed to through biologically relevant processes) social structure through space and time ¹¹.” – see lines 75-76 of the appendix.

L121: Variance should be relatively high.

Response: fixed.

L141: This info seems to be the other way around.

Response: we have changed to read: “We extracted resource selection coefficients for each individual-by-season-by-year and compared them to habitat specialization measures for the same time period.” – see lines 140-141 of the appendix.

L145: ‘specialists’ — but this is conditioned on high vs low fitness.

Response: we have changed to specialist.

L206: It would be more informative if the measures of variance are labelled. It seems only a few individuals exhibited large variation in the left panel.

Response: fixed as suggested

References:

1. Gaillard, J.-M., Festa-bianchet, M., Delorme, D. & Jorgenson, J. Body mass and individual fitness in female ungulates: bigger is not always better. *Proceedings of the Royal Society B* 471–477 (2000).
2. Gaillard, J. M., Yoccoz, N. G., Loison, A. & To, C. Components and Population Dynamics of Large Herbivores. *Annual Review of Ecological Systems* **31**, 367–393 (2000).
3. Gaillard, J. M., Festa-Bianchet, M. & Yoccoz, N. G. Population dynamics of large herbivores: Variable recruitment with constant adult survival. *Trends in Ecology and Evolution* **13**, 58–63 (1998).
4. Silk, J. B. The adaptive value of sociality in mammalian groups. *Philosophical Transactions of the Royal Society B* **362**, 539–559 (2007).
5. Snyder-Mackler, N. *et al.* Social determinants of health and survival in humans and other animals. *Science* **368**, eaxx9553 (2020).
6. Bonar, M. *et al.* Geometry of the ideal free distribution: individual behavioural variation and annual reproductive success in aggregations of a social ungulate. *Ecology Letters* **23**, 1360–1369 (2020).
7. Fretwell, S. D. & Lucas, H. L. J. On territorial behaviour and other factors influencing habitat distribution in birds. *Acta Biotheoretica* **19**, 16–36 (1969).
8. Harris, N. C., Kauffman, M. J. & Mills, L. S. Inferences About Ungulate Population Dynamics Derived From Age Ratios. *Journal of Wildlife Management* **72**, 1143–1151 (2010).
9. Virgl, J. A., Rettie, W. J. & Coulton, D. W. Spatial and temporal changes in seasonal range attributes in a declining barren-ground caribou herd. *Rangifer* **37**, 31 (2017).

10. Festa-Bianchet, M., Gaillard, J.-M. & Jorgenson, J. T. Mass- and density-dependent reproductive success and reproductive costs in a capital breeder. *American Naturalist* **152**, 367–379 (1998).
11. Farine, D. R. A guide to null models for animal social network analysis. *Methods in Ecology and Evolution* **8**, 1309–1320 (2017).